# Performance of real-time PCR and immunofluorescence assay for diagnosis of *Pneumocystis* pneumonia in real-world clinical practice

**Darunee Chotiprasitsakul**[1]*, **Pataraporn Pewloungsawat**[2], **Chavachol Setthaudom**[3], **Pitak Santanirand**[3], **Prapaporn Pornsuriyasak**[4]

1 Division of Infectious Diseases, Department of Medicine, Faculty of Medicine Ramathibodi Hospital, Mahidol University, Bangkok, Thailand, 2 Department of Medicine, Faculty of Medicine Ramathibodi Hospital, Mahidol University, Bangkok, Thailand, 3 Department of Pathology, Faculty of Medicine Ramathibodi Hospital, Mahidol University, Bangkok, Thailand, 4 Division of Pulmonary and Critical Care, Department of Medicine, Faculty of Medicine Ramathibodi Hospital, Mahidol University, Bangkok, Thailand

* darunee.cho@mahidol.ac.th

**Data Availability Statement:** The dataset used in this manuscript is available from the Open Science Forum via the weblink: https://osf.io/3cgh4.

## Abstract

### Background

PCR is more sensitive than immunofluorescence assay (IFA) for detection of *Pneumocystis jirovecii*. However, PCR cannot always distinguish infection from colonization. This study aimed to compare the performance of real-time PCR and IFA for diagnosis of *P. jirovecii* pneumonia (PJP) in a real-world clinical setting.

### Methods

A retrospective cohort study was conducted at a 1,300-bed hospital between April 2017 and December 2018. Patients whose respiratory sample (bronchoalveolar lavage or sputum) were tested by both *Pneumocystis* PCR and IFA were included. Diagnosis of PJP was classified based on multicomponent criteria. Sensitivity, specificity, 95% confidence intervals (CI), and Cohen's kappa coefficient were calculated.

### Results

There were 222 eligible patients. The sensitivity and specificity of PCR was 91.9% (95% CI, 84.0%–96.7%) and 89.7% (95% CI, 83.3%–94.3%), respectively. The sensitivity and specificity of IFA was 7.0% (95% CI, 2.6%–14.6%) and 99.2% (95% CI, 95.6%–100.0%), respectively. The percent agreement between PCR and IFA was 56.7% (Cohen's kappa -0.02). Among discordant PCR-positive and IFA-negative samples, 78% were collected after PJP treatment. Clinical management would have changed in 14% of patients using diagnostic information, mainly based on PCR results.

**Funding:** The authors received no specific funding for this work.

**Competing interests:** The authors have declared that no competing interests exist.

## Conclusions

PCR is highly sensitive compared with IFA for detection of PJP. Combining clinical, and radiological features with PCR is useful for diagnosis of PJP, particularly when respiratory specimens cannot be promptly collected before initiation of PJP treatment.

## Introduction

*Pneumocystis jirovecii* pneumonia (PJP) is a life-threatening lung infection in immunocompromised patients [1] and a common opportunistic infection in human immunodeficiency virus (HIV) patients with CD4 counts $< 200$ cells/mm$^3$ [2]. The typical presentation of PJP in HIV patients consists of a well-known triad of dyspnea, fever, and cough [3]. PJP is gaining more attention in HIV-negative immunocompromised patients, including solid organ transplant recipients, patients with hematologic or solid malignancies, and patients treated with corticosteroids or other immunosuppressive drugs for connective tissue or chronic inflammatory diseases. These patients can develop PJP with less specific symptoms, which carries a poorer prognosis despite early definitive treatment [3–7]. Microscopic examination to detect trophic forms or cysts using staining methods, such as May-Grünwald-Giemsa and Gomori-Grocott methenamine-silver (GMS), or immunofluorescence assays (IFAs) has been the gold standard for diagnosis of PJP [8]. However, these techniques require skill and experience. Microscopic diagnosis may be falsely negative, particularly in patients with low fungal burdens [9].Various PCR targets and methods have been developed and are now widely used in clinical practice. PCR has a sensitivity of 94%–100% and a specificity of 79%–96% for diagnosis of microscopically-positive PJP [8]. Although the high sensitivity of PCR allows for diagnosis of patients with low fungal loads, it can also lead to overdiagnosis of PJP in patients with colonization and not infection [9]. Transient colonization has been observed in HIV patients, non-HIV patients, and healthy subjects. *Pneumocystis* colonization may be linked to the development or transmission of disease [10].

Most laboratories perform their own individually standardized *Pneumocystis* PCR. The accuracy of commercial PCR assays is usually compared with IFA or other PCR assays. Thus, the comparability of the various available PCR methods is unclear.

The objective of this study was to evaluate the performance of real-time PCR and IFA for diagnosis of *Pneumocystis* pneumonia in real-world clinical practice.

## Materials and methods

### Study population

This retrospective cohort study was conducted at Ramathibodi Hospital, a 1,300-bed tertiary care hospital in Bangkok, Thailand, between April 2017 and December 2018. The database was assessed on 6 June 2018 and 20 Nov 2019. All patients aged >18 years whose respiratory samples (bronchoalveolar (BAL) fluid or sputum) were concomitantly tested for PJP by real-time PCR and IFA were included in the study. Patients who had no test results, no chest radiograph, or whose medical records were unavailable were excluded. Patients with a diagnosis of PJP were classified as definite cases and potential cases based on clinical symptoms and signs, radiological findings, histological examination, and response to treatment.

This study was approved by the Ethics Committee of the Faculty of Medicine Ramathibodi Hospital, Mahidol University, with a waiver of informed consent.

## Data collection

Medical records were retrospectively reviewed. Demographic and clinical data were collected including age, sex, immunosuppressive risk factors (HIV status, malignancy, autoimmune diseases, organ transplantation, corticosteroid therapy, diabetes mellitus, and end-stage renal disease), history of PJP prophylaxis, other infections at the time of PJP testing (bacterial, viral, or fungal), radiographic findings, PJP treatment, results of real-time PCR and IFA for PJP, cytology, and pathology.

Sputum was obtained by expectoration, or tracheal suction. BAL fluid was collected via bronchoscopy, as previously described [11]. Instillation of normal saline 50 ml, suction with low pressure, and expulsion into glass bottles was repeatedly done 2–4 times.

## Immunofluorescence assays

Fluorescent antibody staining using monoclonal mouse anti-*P. jirovecii* antibody (clone 3F6; Dako ApS, Glostrup, Denmark) was conducted. The samples were mixed with 0.25% trypsin in phosphate-buffered saline, incubated at room temperature for 5 min, and sedimented. Indirect immunofluorescence on acetone-fixed smears was performed using monoclonal mouse anti- *P. jirovecii*, followed by polyclonal rabbit anti-mouse immunoglogulins/FITC in accordance with the manufacturer's instructions. Trained, experienced personnel examined the slides under the microscope at 20× and 40× magnification for any fluorescent antibody-antigen complexes. Results were given qualitatively. If one or more *P. jirovecii* cysts or trophozoites with typical morphology were observed, the specimen was interpreted as positive. The turnaround time for IFA was 3 days.

## Qualitative and quantitative real-time PCR

DNA extraction from respiratory samples was performed using the magLEAD® 12gC analyzer (Precision System Science, Matsudo, Japan), in accordance with the manufacturer's instructions. Real-time PCR was carried out using the RIDA® GENE *P. jirovecii* assay (R-Biopharm AG, Darmstadt, Germany) in accordance with the manufacturer's instructions. This was a qualitative and quantitative assay targeting the mitochondrial large subunit (mtLSU) rRNA of *P. jirovecii.* Negative controls and positive controls were included in each assay. The PCR results were evaluated via cycle thresholds ($C_T$). The cycle number at which the fluorescence generated within a reaction crossed the threshold was recorded. High concentrations of *P. jirovecii* DNA were associated with low $C_T$ values and vice-versa. A $C_T$ value of 45 was the limit of the detection of this assay. In accordance with the manufacturer's instructions, a $C_T$ value $> 40$ excluded PJP. The turnaround time for PCR was 3 days.

## Cytology

BAL fluid was processed using liquid-based slide systems for ThinPrep (Hologic Inc., Mariborough, MA, USA). The slides were prepared following the manufacturer's instructions. If granular cast or foamy material or honeycomb appearance was seen in the Papanicolaou-stained smear, then a GMS stain was performed to confirm the presence of cyst forms of *Pneumocystis.*

## Pathology

Lung biopsy sample was taken via bronchoscopy [11]. A total of 4–6 samples were placed into a glass bottle containing 10% formalin solution, and processed as previously described [12]. All samples were evaluated for the presence of lung tissue, and pathological changes. If clinical,

radiological, or histological findings of routine haematoxylin and eosin staining was suspicious for PJP, then GMS stain was done to detect *Pneumocystis* in tissue sections.

## Classification of patients

Patients were classified according to specified criteria. A definite diagnosis of PJP was established when patients had histologically confirmed PJP or if cytology found foamy materials containing an organism morphologically consistent with *P. jirovecii*. Alternatively, patients had to meet all of the following criteria: (i) presence of at least two of fever, cough, and dyspnea, (ii) compatible findings from radiographic chest imaging, (iii) good response to PJP treatment with trimethoprim-sulfamethoxazole or alternative antibiotics, and (iv) no other potential etiology to explain respiratory abnormalities. Potential PJP was defined by (i) presence of at least two of fever, cough, and dyspnea, (ii) compatible findings from radiographic chest imaging, and (iii) other concurrent etiology simultaneously causing respiratory abnormalities. Patients were independently adjudicated by two physicians (DC and PaP). Disagreements in adjudication were resolved via a consensus discussion with 1–2 pulmonary physicians.

## Statistical analysis

Medians (interquartile range, IQR) and frequencies (%) were calculated for baseline characteristics and results of laboratory investigations. Chi-square and Wilcoxon rank sum tests were used to assess differences between categorical and continuous variables, respectively. The sensitivity, specificity, and 95% confidence interval (95% CI) of real-time PCR and IFA for PJP were calculated. Percent agreement and Cohen's kappa coefficient were calculated. Values of $P < 0.05$ were considered statistically significant. Data analysis was performed using Stata software, version 15.1 (Stata Corp., College Station, TX, USA).

## Results

A total of 406 patients had their respiratory samples tested for PJP between April 2017 and December 2018. Ninety-four patients were excluded; 37 had samples submitted from outside hospitals, 56 were duplicate samples, and one patient's medical records were destroyed. Among the remaining patients, 80 samples were tested only by IFA and 10 were tested only by real-time PCR. A total of 222 patients whose samples were concomitantly tested by IFA and real-time PCR were included in the analysis. Eighty-six patients (38.7%) had PJP; 43 had definite PJP and 43 had potential PJP. Among the 148 patients from whom BAL fluid samples were collected, 56 (37.8%) were diagnosed with PJP. Among the 74 patients from whom sputum was collected, 30 (40.5%) were diagnosed with PJP (Fig 1). A comparison of baseline characteristics between PJP and non-PJP patients is shown in Table 1. The median ages of the PJP and non-PJP groups were not significantly different at 57 years (IQR, 41–68 years) and 58 years (IQR, 43–68 years), respectively. There were more patients with HIV in the PJP group than the non-PJP group (27.9% versus 9.6%, $P < 0.001$). By contrast, the PJP group had fewer patients with hematologic malignancies than the non-PJP group (16.3% versus 33.1%, $P = 0.006$). The proportions of patients receiving PJP prophylaxis in the PJP and non-PJP groups were 5.8% and 27.2%, respectively ($P < 0.001$). There was a lower proportion of fungal infection in the PJP group than in the non-PJP group (5.8% versus 18.4%, $P = 0.008$). All PJP patients received PJP therapy and 64.0% received steroids as part of PJP treatment regimens.

The most common radiographic findings in the PJP group were ground glass opacities (39.5%), reticular opacities (29.1%), and reticulonodular opacities (23.3%). By contrast, the

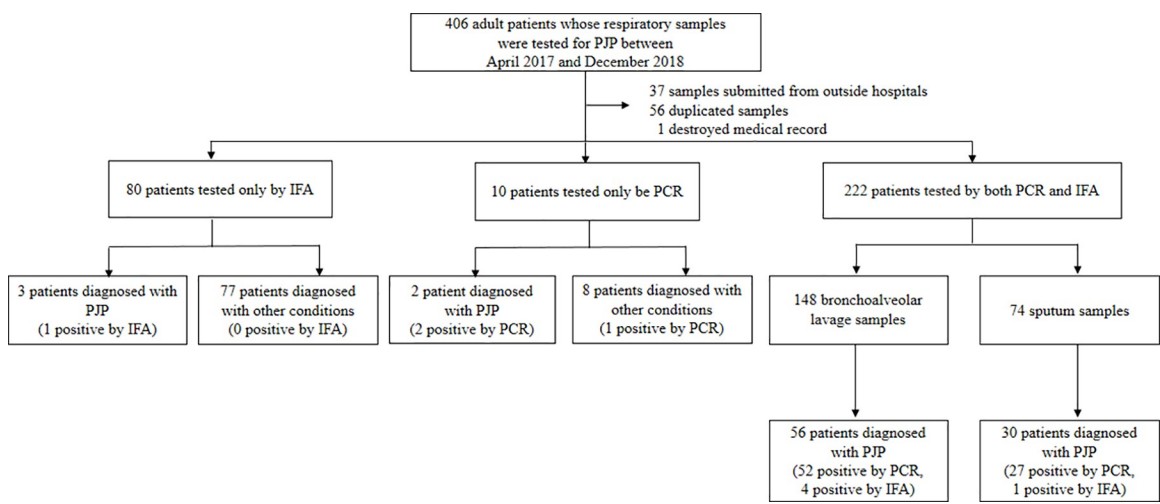

**Fig 1. Study enrollment and results of patients whose respiratory samples were tested.** PJP, *Pneumocystis jirovecii* pneumonia; PCR, polymerase chain reaction; IFA, immunofluorescence assay.

most common radiographic findings in the non-PJP group were reticular opacities (23.5%), reticulonodular opacities (22.8%), and mixed patterns (17.7%) (Table 2).

Of the 86 PJP patients, 79 (91.9%) had positive real-time PCR results and six (7.0%) had positive IFA results. Of the 136 non-PJP patients, 14 (10.3%) had positive real-time PCR results and one (0.7%) had a positive IFA result. The non-PJP patient with a positive IFA result was diagnosed with radiation pneumonitis and did not receive PJP treatment. Her condition clinically improved following steroid treatment. Among the PJP group, presence of foamy exudates was detected in 25 (36.2%) of 69 cytology samples, and PJP was identified in two of 23 pathology samples (8.7%) (Table 2).

Overall, the sensitivity and specificity of real-time PCR for PJP diagnosis was 91.86% (95% CI, 83.95%–96.66%) and 89.71% (95% CI, 83.33%–94.26%), respectively. The sensitivity of real-time PCR was 91.67% (95% CI, 73.00%–98.97%) among HIV patients and 91.94% (95% CI, 82.71%–97.73%) among non-HIV patients. The sensitivity and specificity of real-time PCR for PJP diagnosis from BAL fluid was 92.86% (95% CI, 82.71%–98.02%), and 91.30% (95% CI, 83.58%–96.17%), respectively. The sensitivity and specificity of real-time PCR for PJP diagnosis from sputum was 90.00% (95 CI, 73.47%–97.89%) and 91.30% (95% CI, 83.58%–96.17%), respectively. The overall sensitivity and specificity of IFA for PJP diagnosis was 6.98% (95% CI, 2.60%–14.57%) and 99.19% (95% CI, 95.55%–99.98%), respectively (Table 3). The percent agreement between real-time PCR and IFA was 56.67% (Cohen's kappa -0.02). Of the 114 patients receiving PJP treatment, 81 (71.1%) received treatment prior to sample collection. Seventy-one of 91 discordant PCR-positive and IFA-negative samples (78.0%) in our study were collected following PJP treatment (Table 4).

Of the 222 patients, 108 (48.6%) received empirical PJP treatment prior to PCR and IFA results being known. Six patients were treated for PJP after the PCR and IFA results were available. The PCR or IFA results would have changed the management of 31 patients (14.0%) (Fig 2).

## Discussion

Prompt diagnosis and early treatment of PJP is crucial for optimal patient outcomes. High sensitivity tests are important to identify serious but treatable diseases. A patient who has a

**Table 1. Baseline characteristics of PJP and non-PJP patients.**

| | PJP patients | | | Non-PJP patients | | | P value |
|---|---|---|---|---|---|---|---|
| | Total (N = 86) | Non-HIV (N = 62) | HIV[a] (N = 24) | Total (N = 136) | Non-HIV (N = 123) | HIV[a] (N = 13) | |
| Age, years | 57 (41–68) | 61 (53–71) | 42 (37–51) | 58 (43–68) | 60 (45–69) | 43 (40–54) | 0.91 |
| Male | 39 (45.4) | 33 (53.2) | 14 (58.3) | 73 (53.7) | 56 (45.5) | 7 (53.9) | 0.23 |
| **Pre-existing medical conditions** | | | | | | | |
| CD4 count, cells/mm$^3$ | 33 (8–54) | - | 33 (8–54) | 47 (19–138) | - | 47 (19–138) | 0.13 |
| Hematologic malignancy | 14 (16.3) | 14 (22.6) | 0 (0) | 45 (33.1) | 43 (35.0) | 2 (15.4) | 0.006 |
| Solid malignancy | 8 (9.3) | 8 (12.9) | 0 (0) | 20 (14.7) | 20 (16.3) | 0 (0) | 0.24 |
| Bone marrow transplantation | 3 (3.5) | 3 (4.8) | 0 (0) | 10 (7.4) | 10 (8.1) | 0 (0) | 0.23 |
| Kidney transplantation | 12 (14.0) | 12 (19.4) | 0 (0) | 21 (15.4) | 21 (17.1) | 0 (0) | 0.76 |
| Diabetes | 14 (16.3) | 12 (19.4) | 2 (8.3) | 26 (19.1) | 24 (19.5) | 2 (15.4) | 0.59 |
| Chronic kidney disease | 12 (14.0) | 12 (19.4) | 0 (0) | 23 (16.9) | 23 (18.7) | 0 (0) | 0.56 |
| Connective tissue diseases | 22 (25.6) | 20 (32.3) | 2 (8.3) | 30 (22.1) | 29 (23.6) | 0 (0) | 0.55 |
| Corticosteroid treatment for ≥14 day | | | | | | | 0.60 |
| ≤15 mg/day prednisolone | 24 (27.9) | 23 (37.1) | 1 (4.2) | 31 (22.8) | 31 (25.2) | 0 (0) | |
| >15 mg/day prednisolone | 19 (22.1) | 18 (29.0) | 1 (4.2) | 28 (20.6) | 26 (21.1) | 2 (15.4) | |
| PJP prophylaxis | 5 (5.8) | 3 (4.8) | 2 (8.3) | 37 (27.2) | 31 (25.2) | 6 (46.2) | <0.001 |
| **Symptoms** | | | | | | | |
| Fever | 69 (80.2) | 48 (77.4) | 21 (87.5) | 93 (68.4) | 82 (66.7) | 11 (84.6) | 0.05 |
| Dyspnea | 78 (90.7) | 57 (91.9) | 21 (87.5) | 93 (68.4) | 85 (69.1) | 8 (61.5) | <0.001 |
| Non-productive cough | 36 (41.9) | 27 (43.6) | 9 (37.5) | 22 (16.2) | 18 (14.6) | 4 (30.8) | <0.001 |
| Productive cough | 30 (34.9) | 19 (30.7) | 11 (45.8) | 56 (41.2) | 50 (40.7) | 6 (46.2) | 0.35 |
| **Other infections** | | | | | | | |
| Bacteria | 18 (20.9) | 9 (14.5) | 9 (37.5) | 46 (33.8) | 40 (32.5) | 6 (46.2) | 0.04 |
| Virus | 20 (23.3) | 13 (21.0) | 7 (29.2) | 27 (19.9) | 25 (20.3) | 2 (15.4) | 0.55 |
| Fungus | 5 (5.8) | 3 (4.8) | 2 (8.3) | 25 (18.4) | 21 (17.1) | 4 (30.8) | 0.008 |
| Tuberculosis | 2 (2.4) | 1 (1.7) | 1 (4.2) | 7 (5.2) | 6 (4.9) | 1 (76.7) | 0.31 |
| PJP treatment | 86 (100.0) | 62 (100.0) | 24 (100.0) | 28 (20.6) | 26 (21.1) | 2 (15.4) | <0.001 |
| Time from sample collection to initiation of PJP treatment, days | 1 (0–3) | 1 (1–3) | 1 (0–4) | 1 (0–3) | 1 (0–3) | -8 (-18-3) | 0.47 |
| Receiving steroids for PJP treatment (N = 112) | 55 (64.0) | 37 (59.7) | 18 (75.0) | 13 (46.4) | 11 (42.3) | 2 (100) | 0.10 |

Data are presented as N (%) or median (IQR).

P value for difference between total PJP patients and total non-PJP patients.

[a]P value <0.001 for difference of HIV between total PJP patients and total non-PJP patients.

negative test thus would be less likely to have the true infection [10]. Our study found an extremely low sensitivity of IFA for detection of PJP (6.98%). The sensitivity of IFA in a meta-analysis of diagnostic tests for PJP among HIV-infected patients was 67.1 [13]. However, the sensitivity of IFA was calculated by comparison of induced sputum with BAL fluid [13, 14]. The gold standard is the best available test at a given time. When enough data become available, the gold standard can change [15]. When there is no perfect gold standard test, one approach is to use a composite reference standard. The combination of several imperfect references can yield a better reference standard, reducing bias [16]. Our study defined a reference standard using several clinical, radiographic and laboratory criteria to improve the accuracy of diagnosis and to exclude colonization.

In a previous study of 305 induced sputum specimens among HIV-positive patients, *P. jirovecii* was detected by IFA and PCR in 51 and 67 of specimens, respectively [17]. Our study

**Table 2. Laboratory findings in PJP and non-PJP patients.**

|  | PJP patients (N = 86) | Non-PJP patients (N = 136) | *P* value |
|---|---|---|---|
| Chest radiographic findings |  |  | <0.001 |
| Ground glass opacities | 34 (39.5) | 19 (14.0) |  |
| Reticular opacities | 25 (29.1) | 32 (23.5) |  |
| Reticulonodular opacities | 20 (23.3) | 31 (22.8) |  |
| Consolidation | 3 (3.5) | 7 (5.2) |  |
| Alveolar opacities | 3 (3.5) | 23 (16.9) |  |
| Mixed patterns | 1 (1.2) | 24 (17.7) |  |
| Type of sample |  |  | 0.70 |
| BAL fluid | 56 (65.1) | 92 (67.7) |  |
| Sputum | 30 (34.9) | 44 (32.4) |  |
| Positive PCR for PJP | 79 (91.9) | 14 (10.3) | <0.001 |
| Positive IFA for PJP | 6 (7.0) | 1 (0.7) | 0.01 |
| Cytology (N = 172) | 69 | 103 |  |
| Presence of foamy exudates | 25 (36.2) | 0 (0) | <0.001 |
| Pathology (N = 67) | 23 | 44 |  |
| Positive | 2 (8.7) | 0 (0) | 0.05 |

All data presented as N (%).

found a lower rate of IFA positivity (7.0%) and a higher rate of PCR positivity (91.9%) among 222 BAL fluid and sputum specimens. Potential explanations for the few positive IFA results in our study could be the time of, and process for, specimen collection. When a definitive diagnosis cannot be made in time because of inability to obtain necessary specimens or low fungal burden, empirical treatment decisions are usually made based on a presumptive diagnosis. The basis of this approach is to consider clinical features, radiographic findings and the patient's risk for PJP. The proportion of patients who received treatment prior to sample collection was highest among the discordant PCR-positive IFA-negative group (78.0%). A follow-up biopsy after clinical response to trimethoprim-sulfamethoxazole treatment failed to detect *P. jiroveci* cysts, while trophozoites were still observed [18, 19]. IFA also requires technical expertise, particularly when fungal loads are low. Only one discordant result between IFA and PCR results from BAL samples was observed in a study of 62 HIV patients. IFA could continue to be used for diagnosis of PJP in HIV patients, considering its high agreement with PCR,

**Table 3. Sensitivity and specificity of IFA and PCR for diagnosis of PJP.**

|  | Sensitivity (95% CI) | Specificity (95% CI) |
|---|---|---|
| PCR (N = 222) | 91.86% (83.95%–96.66%) | 89.71% (83.33%–94.26%) |
| HIV patients (N = 37) | 91.67% (73.00%–98.97%) | 76.92% (46.19%–94.96%) |
| Non-HIV patients (N = 185) | 91.94% (82.17%–97.33%) | 91.06% (84.56%–95.45%) |
| BAL fluid (N = 148) | 92.86% (82.71%–98.02%) | 91.30% (83.58%–96.17%) |
| Sputum (N = 74) | 90.00% (73.47%–97.89%) | 86.36% (72.65%–94.83%) |
| IFA (N = 222) | 6.98% (2.60%–14.57%) | 99.19% (95.55%–99.98%) |
| HIV patients (N = 37) | 4.17% (0.11%–21.12%) | 100.00% (75.29%–100.00%) |
| Non-HIV patients (N = 185) | 8.06% (2.67%–17.83%) | 99.19% (95.55%–99.98%) |
| BAL fluid (N = 148) | 7.14% (1.98%–17.29%) | 98.91% (94.09%–99.97%) |
| Sputum (N = 74) | 6.67% (0.82%–22.07%) | 100.00% (91.96%–100.00%) |

Table 4. Agreement of PCR and IFA for diagnosis of PJP.

|  | Positive PCR | Negative PCR |
| --- | --- | --- |
| Positive IFA | 2 (0, 0%) | 5 (2, 40.0%) |
| Negative IFA | 91 (71, 78.0%) | 124 (22, 17.7%) |

Percent agreement: 56.67%, Cohen's k: -0.020

(number, % of patients receiving PJP treatment prior to sample collection); *P* value = 0.01

reasonable cost and rapid processing time [20]. However, a study of 46 specimens from HIV and non-HIV patients found that only 4.3% were positive by IFA, whereas simple PCR and nested PCR were positive in 23.9% and 45.6% of specimens, respectively. This result supports using PCR in patients with clinically suspected PJP when IFA is negative [21]. Furthermore, variation between IFA results in different studies stresses the importance of evaluating test performance in each individual setting.

A meta-analysis of PCR for diagnosis of PJP showed a sensitivity of 0.99 (95% CI, 0.96–1.00) and a specificity of 0.90 (95% CI, 0.87–0.93); however, significant heterogeneity in all diagnostic parameters was observed. The reasons for this heterogeneity included variation in DNA extraction and cell wall disruption processes, different targeted gene regions, and use of quantitative versus qualitative PCR [22]. The sensitivity and specificity of mtLSU rRNA PCR, using IFA as the gold standard, were 94.6% and 89.1%, respectively [17]. Another study showed that the sensitivity and specificity of PCR targeting the same gene region in comparison with IFA were 78.0% and 100%, respectively [19]. The sensitivity and specificity of PCR targeting this gene region in the present study, using a composite criterion, were 91.86% and 89.71%, respectively. The high sensitivity of PCR is useful for diagnosis of patients with low fungal burdens. However, *P. jirovecii* may be a dynamic reservoir, with transient asymptomatic colonization in immunocompetent subjects or those in close contact with infected patients

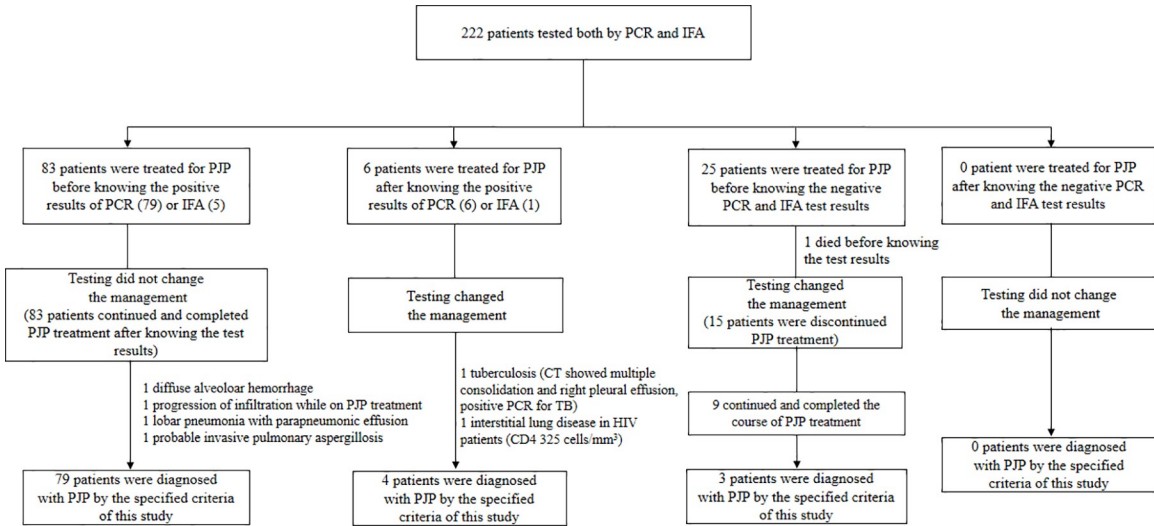

Fig 2. Treatment of patients whose respiratory samples were tested for PJP by PCR and IFA. Specified criteria: Definite PJP (i) presence of at least two of fever, cough, and dyspnea, (ii) compatible radiographic chest imaging, (iii) good response to PJP treatment, and (iv) no other potential etiology. Potential PJP (i) presence of at least two of fever, cough, and dyspnea, (ii) compatible radiographic chest imaging, and (iii) other concurrent etiology causing respiratory abnormalities. PJP, *Pneumocystis jirovecii* pneumonia; PCR, polymerase chain reaction; IFA, immunofluorescence assay.

[23, 24]. Thus, PCR might lead to overdiagnosis of PJP in patients with colonization [9]. Interpretation of positive PCR results when microscopic staining is undetectable is difficult [10]. Our study focused on patients with clinical infection. Employing different definitions of diagnosis could lead to different classifications of disease, and subsequently to different results of sensitivity and specificity for PCR. The sensitivity of PCR among HIV and non-HIV patients in our study was 91.67%, and 91.94%, respectively. Lower sensitivity of PCR, using clinical and radiographic findings as a gold standard, was observed in non-HIV patients compared with all patients (52% versus 65%) [25]. Difficulties in PJP diagnosis in non-HIV immunocompromised patients were observed in several studies [9, 26]. The $C_T$ cut-off value for a positive PCR result in non-HIV-infected patients varied from 22 to 35 [27, 28]. A higher $C_T$ cut-off value of 40 was used in this study in accordance with the manufacturer's instructions and might result in higher sensitivity among non-HIV patients. The higher $C_T$ cut-off value would increase the chance of detecting *Pneumocystis*, as well as false-positives from artifact or cross contamination. Although a strongly positive PCR ($C_T$ <31.5) was highly suggestive of PJP, a weakly positive PCR ($C_T$ 31.5–45) could be colonization or PJP undergoing treatment [29]. Therefore, PCR result should be carefully interpreted in association with clinical data to discriminate active infection from colonization, and false-positives. Although only two samples had a false negative PCR result among HIV patients, lower specificity of PCR among HIV patients may have been influenced by the small number of HIV patients in our study.

In our study, the sensitivity and specificity of PCR from sputum samples were both slightly lower than from BAL samples (90.00% versus 92.86%, and 86.36% versus 91.30%, respectively). PCR was compared with IFA in 120 induced sputum specimens and 112 BAL specimens from HIV and other immunocompromised patients. The sensitivity and specificity of PCR from induced sputum samples were lower than from BAL samples (94% versus 100%, and 90% versus 98%, respectively) [30]. The lower diagnostic performance using sputum may be because these specimens are gathered from the central proximal airway, whereas BAL specimens are gathered from the peripheral airways and the alveolar compartment. Moreover, sputum cannot always be expectorated. Induction with aerosolized hypertonic saline was used to increase the success of sputum expectoration [31]. The fungal burden also correlates with diagnostic performance [32].

Empirical PJP treatment is a common practice for patients with clinically suspected PJP. Of 114 patients receiving PJP treatment, 108 (94.7%) received the treatment before knowing PCR or IFA results. The results of these tests would have led to a change in clinical management for 14% of patients (discontinuation of PJP treatment in 25 patients with negative tests and initiation of PJP treatment in 6 patients with positive tests). The diagnostic value of these tests for clinical management was mainly driven by PCR results.

In this study, we used multiple diagnosis criteria, independent of both PCR and IFA, to reduce incorporation bias and to ascertain true *Pneumocystis* infection. We demonstrated that PCR has higher sensitivity and specificity than IFA for diagnosis of PJP from both BAL and sputum samples in both HIV and non-HIV patients. This result supports using PCR alone instead of IFA alone, or concomitant use of both IFA and PCR, for diagnosis of PJP in clinical practice. Nonetheless, interpretation of PCR results and treatment decisions should be based on a multicomponent algorithm consisting of clinical, radiological parameters and PCR results [9].

A limitation of our study was that the data were retrospectively analyzed. Although both IFA and PCR for suspected cases of PJP were recommended in our routine practice, 90 patients had only IFA or PCR tests, and were thus excluded. However, the numbers of positive tests and true-positive cases were very low. Theoretically, when these cases are included, the effects on sensitivity and sensitivity of the PCR and IFA would be very small (<1%). Review of

cases occurred retrospectively and there may have been clinical data missing from the medical records of patients for whom diagnosis of PJP could not be confidently made. Classification of diagnosis relied on this review and may have resulted in misclassification bias. However, the relevant criteria were defined to achieve homogenous classification, and true infection was independently adjudicated. In uncertain cases, classification was achieved after reaching a consensus.

In conclusion, real-time PCR is highly sensitive compared with IFA for diagnosis of PJP. PCR is a more useful diagnostic tool in settings where prompt specimen collection is not possible prior to initiation of PJP treatment. The performance of PCR should be evaluated in each individual setting, as results might vary depending on PCR methods and the reference gold standard.

## Acknowledgments

We thank Dr. Araya Sukprapruet and Dr. Phichaya Chamnanvej, pulmonary physicians of Somdech Phra Debaratana Medical Center, Faculty of Medicine Ramathibodi Hospital, Mahidol University, for their consultations on consensus decision-making.

## Author Contributions

**Conceptualization:** Darunee Chotiprasitsakul, Pataraporn Pewloungsawat.

**Data curation:** Darunee Chotiprasitsakul, Chavachol Setthaudom, Pitak Santanirand.

**Formal analysis:** Darunee Chotiprasitsakul.

**Investigation:** Darunee Chotiprasitsakul, Pataraporn Pewloungsawat.

**Methodology:** Darunee Chotiprasitsakul, Chavachol Setthaudom, Pitak Santanirand.

**Writing – original draft:** Darunee Chotiprasitsakul, Pataraporn Pewloungsawat.

**Writing – review & editing:** Prapaporn Pornsuriyasak.

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
