## [Decision Letter · Decision Letter 0]

28 Oct 2020

PONE-D-20-27498

Performance of real-time PCR and immunofluorescence assay for diagnosis of Pneumocystis pneumonia in real-world clinical practice

PLOS ONE

Dear Dr. Chotiprasitsakul,

Thank you for submitting your manuscript to PLOS ONE. After careful consideration, we feel that it has merit but does not fully meet PLOS ONE’s publication criteria as it currently stands. Therefore, we invite you to submit a revised version of the manuscript that addresses the points raised during the review process.

We look forward to receiving your revised manuscript.

Kind regards,

Giuseppe Vittorio De Socio, MD, PhD

Academic Editor

PLOS ONE

Journal Requirements:

2. PLOS ONE requires experimental methods to be described in enough detail to allow suitably skilled investigators to fully replicate and evaluate your study. See https://journals.plos.org/plosone/s/submission-guidelines#loc-materials-and-methods for more information.

To comply with PLOS ONE submission guidelines, in your Methods section, please provide a more detailed description of your IFA methodology. Specifically please give a brief overviewof the method and ensure that the names, sources, catalog numbers and dilutions of primary and secondary antibodies are described in your methods.

3. Please include the date(s) on which you accessed the databases or records to obtain the data used in your study.

Reviewers' comments:

Reviewer's Responses to Questions

**Comments to the Author**

1. Is the manuscript technically sound, and do the data support the conclusions?

Reviewer #1: Yes

2. Has the statistical analysis been performed appropriately and rigorously? 

Reviewer #1: Yes

3. Have the authors made all data underlying the findings in their manuscript fully available?

Reviewer #1: Yes

4. Is the manuscript presented in an intelligible fashion and written in standard English?

Reviewer #1: Yes

5. Review Comments to the Author

Reviewer #1: This article is about the comparison of PCR and IFA for the diagnosis of Pneumocystis jirovecii pneumonia (PJP). This an interesting article, however some points need to be clarified.

Introduction:

Line 61 66: Authors should better define the link they are doing between colonization and asymptomatic carriage.

Materials and Methods:

Line 78: I’m surprised that the authors described only two types of samples: BAL and sputum. Are all “BAL” real “BAL”?

How many cycle are performed for the amplification

Line 115 118: What was the purpose of defining CT. In this article, were the CT used for the diagnosis of PJP? It is confusing because in the text there is no mention of the CT to define PJP and authors classified the patients as definite PJP and potential PJP. In addition, as there is no standard for the PCR, it is difficult to compare the CT from on study to another.

Results:

Table 1: Authors should separate HIV and non HIV patients. In additions, when added all the other categories of patients (Hematologic malignancy, solid malignancy…) plus the HIV patients, there are more than 86 patients. Could the authors clarify this classification? This is an important point because immunocompromised HIV patients and immunocompromised non HIV patients are different in terms of diagnosis of PJP.

Were all the data described in table 1 available for all the patients?

Line 173 175: The authors mentioned cytology and pathology examination. Could they describe the technique in the materials and methods section?

Figure 2: I don’t understand the classification “testing did not change the management” For example in the first case, for the 83 patients who were treated before the test results, does the treatment was stopped for those not diagnosed PJP? If so, testing did change the management.

In the second case, 6 patients were treated for PJP after the test result but only 4 were diagnosed PJP. Does this mean that the 2 missing patients had colonization?

Table 3. I would have changed the title as it is not a comparison.

Discussion

Line 212: In the discussion, authors should precise that in the ref 16, the tested population was HIV positive.

Line 219 220: It look like this sentence is a result, but I didn’t find in the result section where it was mentioned. Could the authors clarified this point?

Line 225: PCP? Is it PJP?

Line 246: What does the sentence “Interpretation of PCR…. Is challenging” mean?

Line 255: The sentence “A higher CT cut off…” is in contradiction with the sentence line 117 “In accordance with…”. Could the authors better explain how they consider if it is a PJP or a colonization according to the CT?

Line 281-282: When did the authors differentiate infection from colonization. In the results section, it is only mentioned PJP.

In the light of the sentence line 286 – 288, I would have completed the conclusion in the abstract in order to make it more attractive.

6. PLOS authors have the option to publish the peer review history of their article (what does this mean?). If published, this will include your full peer review and any attached files.

Reviewer #1: No

---

## [Author Response · Author response to Decision Letter 0]

14 Nov 2020

Dear Editor-in-Chief and Reviewers:

We would like to thank the editors and reviewers for the very helpful and constructive comments, which we indeed appreciate. 

We have revised the format of the manuscript, provided more details of IFA and the dates which we accessed the database in the method section. 

Our ethics committee allows sharing a de-identified data set. We have uploaded this data set on Open Science Forum (https://osf.io/3cgh4).

We have addressed all the comments as shown in the revised manuscript with track changes and believe that they have greatly improved the quality of the manuscript. Please find the summary of responses below. Thank you again.

Best regards,

Darunee Chotiprasitsakul

Reviewer #1 evaluation

Introduction:

Line 61 66: Authors should better define the link they are doing between colonization and asymptomatic carriage.

Thank you for this comment. The notion of colonization or asymptomatic carriage of P. jirovecii has been described when P. jirovecii DNA is detected in patients without any symptoms. (N A Maskell, et al. Thorax. 2003 Jul; 58(7): 594–597.) We have revised the word Line 63:

“asymptomatic carriage” to “colonization”

Materials and Methods:

Line 78: I’m surprised that the authors described only two types of samples: BAL and sputum. Are all “BAL” real “BAL”?

How many cycle are performed for the amplification

Types of respiratory specimen for PJP diagnosis include BAL and sputum sample. (Cruciani M, et al. Eur Respir J. 2002 Oct;20(4):982-9) Fiberoptic bronchoscopy with bronchoalveolar lavage (BAL) was performed for diagnostic indications, according to the standard guidelines. The induction of sputum was not performed because of no available negative-pressure room for this procedure at our institution and concern for risk of P. jirovecii airborne transmission. (Valade S, et al. Intensive Care Med. 2015) Sputum was obtained by expectoration, or tracheal suction, which has been shown to provide a comparable quality. (Novaseb V, et al. J Infect Dev Ctries. 2014 Mar 13;8(3):349-57, Choe PG, et al. Med Mycol. 2014 Apr;52(3):326-30) We add more details Line 97-99:

“Sputum was obtained by expectoration, or tracheal suction. BAL fluid was collected via bronchoscopy, as previously described [11]. Instillation of normal saline 50 ml, suction with low pressure, and expulsion into glass bottles was repeatedly done 2-4 times.”

Based on the LOD determinations (Ct ≥40), the PCR amplification war performed until no longer expected to be detected at 45 cycles. Line 121-122:

“A CT value of 45 was the limit of the detection of this assay.”

Line 115 118: What was the purpose of defining CT. In this article, were the CT used for the diagnosis of PJP? It is confusing because in the text there is no mention of the CT to define PJP and authors classified the patients as definite PJP and potential PJP. In addition, as there is no standard for the PCR, it is difficult to compare the CT from on study to another.

Thank you for this comment. We would like to demonstrate the available reference CT cut-off value. We agree that there is no standard cycle threshold and could cause confusion. 

We delete Line 122-123

“A CT value > 35 could exclude Pneumocystis colonization and pneumonia with a specificity of 80% and a sensitivity of 80% [11].”

Results:

Table 1: Authors should separate HIV and non HIV patients. In additions, when added all the other categories of patients (Hematologic malignancy, solid malignancy…) plus the HIV patients, there are more than 86 patients. Could the authors clarify this classification? This is an important point because immunocompromised HIV patients and immunocompromised non HIV patients are different in terms of diagnosis of PJP.

Were all the data described in table 1 available for all the patients?

Thank you for this suggestion and double check the number of co-morbidities. We have revised table 1, divided the total, HIV and non-HIV patients. We have included our de-identified data on Open Science Forum (https://osf.io/3cgh4).

Some patients had more than one active pre-existing conditions as shown in the table below.

Number of pre-existing medical conditions PJP patients

(N=86) non-PJP patients

(N=136)

0 2 patients 5 patients

1 35 patients 58 patients

2 30 patients 41 patients

3 19 patients 21 patients

4 0 patients 11 patients

Total 86 patients 136 patients

Line 173 175: The authors mentioned cytology and pathology examination. Could they describe the technique in the materials and methods section?

Following your suggestions, we have added the description of cytology and pathology Line 127- 139:

“Cytology

BAL fluid was processed using liquid-based slide systems for ThinPrep (Hologic Inc., Mariborough, MA, USA). The slides were prepared following the manufacturer’s instructions. If granular cast or foamy material or honeycomb appearance was seen in the Papanicolaou-stained smear, then a GMS stain was performed to confirm the presence of cyst forms of Pneumocystis.

Pathology

Lung biopsy sample was taken via bronchoscopy [11]. A total of 4-6 samples were placed into a glass bottle containing 10% formalin solution, and processed as previously described [12]. All samples were evaluated for the presence of lung tissue, and pathological changes. If clinical, radiological, or histological findings of routine haematoxylin and eosin staining was suspicious for PJP, then GMS stain was done to detect Pneumocystis in tissue sections.”

Figure 2: I don’t understand the classification “testing did not change the management” For example in the first case, for the 83 patients who were treated before the test results, does the treatment was stopped for those not diagnosed PJP? If so, testing did change the management. 

Thank you for this comment. All 83 patients continued and completed PJP treatment after knowing positive test results. All were diagnosed PJP by attending physicians during admission, but 4 patients did not meet the specified criteria of this study (clinical course, imaging, and treatment response) for diagnosis of PJP. 

In the second case, 6 patients were treated for PJP after the test result but only 4 were diagnosed PJP. Does this mean that the 2 missing patients had colonization?

Yes, the 2 missing patients had colonization.

We have revised and add more details on Figure 2.

Table 3. I would have changed the title as it is not a comparison.

Thank you for this suggestion. We have revised the title of Table 3

“Sensitivity and specificity of IFA and PCR for diagnosis of PJP.”

Discussion

Line 212: In the discussion, authors should precise that in the ref 16, the tested population was HIV positive.

We have revised accordingly. Line 242: 

“In a previous study of 305 induced sputum specimens among HIV-positive patients”

Line 219 220: It look like this sentence is a result, but I didn’t find in the result section where it was mentioned. Could the authors clarified this point?

Thank you for this comment. We have revised the result and discussion section.

Result Line 218-219 “Seventy-one of 91 discordant PCR-positive and IFA-negative samples (78.0%) in our study were collected following PJP treatment (Table 4).”

Discussion Line 252-253 “The proportion of patients who received treatment prior to sample collection was highest among the discordant PCR-positive IFA-negative group (78.0%).”

Line 225: PCP? Is it PJP?

We have corrected the word Line 258: “PJP”

Line 246: What does the sentence “Interpretation of PCR…. Is challenging” mean?

Thank you for this suggestion. We changed the word “challenging” to “difficult” Line 280.

Line 255: The sentence “A higher CT cut off…” is in contradiction with the sentence line 117 “In accordance with…”. Could the authors better explain how they consider if it is a PJP or a colonization according to the CT?

Thank you for this comment. The higher CT allows the amplification of DNA at more cycles than lower CT. Therefore, higher CT cut-off increases the chance of detecting pneumocystis. As the cut-off increases, sensitivity, as well as the false-positives would increase, while false-negatives would decrease. Positive detection by CT could be 1. PJP, 2. colonization, 3. Artifacts/cross contaminations. We have add more explanation Line 290-295:

“The higher CT cut-off value would increase the chance of detecting pneumocystis, as well as false-positives from artifact or cross contamination. Although a strongly positive PCR (CT <31.5) was highly suggestive of PJP, a weakly positive PCR (CT 31.5-45) could be colonization or PJP undergoing treatment.”

Line 281-282: When did the authors differentiate infection from colonization. In the results section, it is only mentioned PJP.

We have revised Line 319-320:

“to ascertain true pneumocystis infection.”

In the light of the sentence line 286 – 288, I would have completed the conclusion in the abstract in order to make it more attractive.

Thank you for this suggestion. We have revised the abstract Line 38-40:

“Combining clinical, and radiological features with PCR is useful for diagnosis of PJP, particularly when respiratory specimens cannot be promptly collected before initiation of PJP treatment.”

---

## [Editor Report · Decision Letter 1]

2 Dec 2020

Performance of real-time PCR and immunofluorescence assay for diagnosis of Pneumocystis pneumonia in real-world clinical practice

PONE-D-20-27498R1

Dear Dr. Chotiprasitsakul,

We’re pleased to inform you that your manuscript has been judged scientifically suitable for publication and will be formally accepted for publication once it meets all outstanding technical requirements.

Kind regards,

Giuseppe Vittorio De Socio, MD, PhD

Academic Editor

PLOS ONE
---

## [Editor Report · Acceptance letter]

10 Dec 2020

PONE-D-20-27498R1 

Performance of real-time PCR and immunofluorescence assay for diagnosis of *Pneumocystis* pneumonia in real-world clinical practice 

Dear Dr. Chotiprasitsakul:

I'm pleased to inform you that your manuscript has been deemed suitable for publication in PLOS ONE. Congratulations! Your manuscript is now with our production department. 

Kind regards, 

on behalf of

Dr. Giuseppe Vittorio De Socio 

Academic Editor

PLOS ONE